# Parents and carers' experiences of seeking health information and support online for long-term physical childhood conditions: a systematic review and thematic synthesis of qualitative research

Bethan Mair Treadgold 🔟 ,[1] Emma Teasdale 🔟 ,[1] Ingrid Muller 🔟 ,[1] Amanda Roberts,[2] Neil Coulson,[3] Miriam Santer[1]

For numbered affiliations see end of article.

**Correspondence to**
Bethan Mair Treadgold;
b.m.treadgold@southampton.ac.uk

## ABSTRACT

**Objective**  To systematically review and synthesise qualitative research exploring parents/carers' experiences of seeking online information and support for long-term physical childhood conditions.

**Design**  Systematic review and thematic synthesis of qualitative research.

**Data sources**  Medline, CINAHL, Embase, PsycINFO and the International Bibliography of the Social Sciences were searched from inception to September 2019. We used thematic synthesis to analyse findings.

**Eligibility criteria**  Primary research papers presenting qualitative data collection and analysis, focusing on parents/carers' experiences of seeking health information and support from online resources for long-term physical childhood health conditions. No language restrictions were placed.

**Results**  23 studies from seven countries met inclusion criteria and were included in the synthesis. Included studies presented data collected through interviews/focus groups with 559 parents/carers; free-text surveys and essays with 26 parents/carers and 2407 messages from online support groups. Parents/carers developed a variety of strategies to obtain information and support online, based on personal preferences, appraisal of trustworthiness, perceived needs and previous experiences online. Many parents sought the benefits of online information and support, which included reassurance and validation from online communities, and feeling they had greater knowledge about their children's conditions. Some concerns and perceived risks were discussed, which often stemmed from prior unsatisfactory experiences of seeking information and support online, consultations with health professionals and seeing distressing stories online.

**Conclusion**  Most parents/carers were successful in obtaining information and support online. Many continued to share experiences with other parents/carers online. The need for information was particularly apparent early after diagnosis of the condition, whereas the need for peer support continued. The potential concerns and perceived risks with information and support online were especially apparent among parents/carers of children with life-limiting long-term conditions. Findings may be useful for

### Strengths and limitations of this study

► The first systematic review to synthesise the existing qualitative literature exploring parents/carers' experiences of seeking health information and support online for long-term physical childhood conditions.

► A comprehensive study synthesising the experiences of hundreds of parents/carers of children with a broad range of long-term physical conditions which provides new insights and develops the research field.

► A systematic and comprehensive search of the literature using multiple databases coupled with detailed synthesis of included studies.

► Excludes the online experiences of parents/carers of children with acute, mental health or developmental conditions and otherwise healthy children.

► Excluded quantitative studies which could have been used to support the qualitative findings.

health professionals to facilitate discussions regarding use of online resources, and researchers designing online health resources for parents/carers.

**PROSPERO registration number**  CRD42018096009.

## BACKGROUND

Caring for a child with a long-term condition can be challenging for parents and carers, requiring ongoing management with regular treatment use, and often complicated, time-consuming and emotionally laden everyday routines.[1] Seeking to maintain treatment and care regimes and dealing with social and financial constraints, while balancing the needs of the whole family, increase these challenges.[2] For the remainder of the paper, parents and carers are collectively referred to as 'parents'.

As a source of health information, the internet is convenient, instant and can help

enhance knowledge and understanding of a condition.[3] As a source of support, the internet also enables parents to exchange stories with other parents with similar experiences. This commonly occurs through asynchronous discussion platforms,[4] although synchronous (real-time) discussions through online platforms are also increasingly popular. Reviews of qualitative research have explored how the internet influences peoples' health experiences.[5 6] Experiences include obtaining health information to understand illness, redressing offline information and knowledge deficits, discussing experiences with health services, building connections with others, feeling supported and validated, releasing emotions and offline frustrations and learning self-management behaviours from others. Other reviews of qualitative research surrounding this area have explored the views of physicians regarding social media interventions to support child health,[7] and experiences with health information online for carers of adults living with cancer.[8] However, no systematic review has been conducted that synthesises the qualitative research on the online experiences of parents of children with long-term physical conditions.

## AIM

This review aims to explore parents' experiences of seeking information and support online for long-term physical childhood conditions.

## METHOD

This study follows the Centre for Reviews and Dissemination's guidance for undertaking reviews in healthcare,[9] and the enhancing transparency in reporting the synthesis of qualitative research (ENTREQ) guidelines for reporting the synthesis of qualitative research.[10]

### Eligibility criteria

We sought studies that explored parents' experiences of using the internet for information and support about long-term physical childhood condition(s) (table 1). To be eligible for inclusion, studies had to present qualitative data collection and analysis methods and qualitative data. Studies that used a mixed methodology were included if they comprised a qualitative component substantive enough to perform a qualitative analysis.

### Search strategy

Five electronic databases were searched (Medline, CINAHL, Embase, PsycINFO and the International Bibliography of the Social Sciences) from the earliest date possible to September 2019. The Cochrane Library of Systematic Reviews and PROSPERO were also searched to check for any ongoing similar reviews. A comprehensive search strategy was developed with a medical librarian from the University of Southampton for each database (table 2). Search terms seeking to identify only qualitative research were initially included in search strategies, but we found that they did not improve the search

| Table 1 | Eligibility criteria | |
|---|---|---|
| | **Inclusion criteria** | **Exclusion criteria** |
| Setting | ► Online health information and support resources (eg, websites, forums, social media pages, blogs) | ► Primarily offline health information and support resources (eg, family, friends, books, leaflets) |
| Design | ► Qualitative data collection methods (or mixed-method studies involving qualitative data) (eg, interviews), qualitative data analysis methods (eg, thematic analysis) and presented qualitative data (eg, themes, narratives and quotations) | ► Did not present the qualitative study criteria |
| | ► Primary research (eg, full research papers, feasibility studies and pilot studies) | ► Not primary research (eg, reviews, protocols and commentaries) |
| Population | ► Parents and carers of children living with long-term physical health conditions | ► Not parents or carers (eg, family members without caring responsibility for the child, health professionals and teachers) |
| Condition | ► Long-term physical health conditions Definition: A long-term or chronic physical health condition is a condition that requires ongoing management which cannot currently be cured but can be controlled with regular adherence to treatment[43] | ► Acute conditions (eg, common cold) |
| | | ► Mental health conditions (eg, depression) |
| | | ► Developmental disorders of learning, language and cognition (eg, attention deficit hyperactivity disorder) |
| | | ► Healthy populations or health-promoting related studies (eg, diet, physical activity, obesity) |
| Outcome | ► Parents and carers' experiences | ► Not parents or carers' experiences (eg, patterns of internet usage derived from quantitative studies) |

**Table 2** Search terms and syntax for each database

| Database | Internet search terms | Parent/carer search terms | Use of online health information search terms | Paediatric search terms |
|---|---|---|---|---|
| MEDLINE | 1. exp Internet/<br>2. forum*.mp.<br>3. facebook.mp.<br>4. social media.mp.<br>5. blog*.mp.<br>6. Internet.mp.<br>7. (web or online).mp. | 1. parents/ or fathers/ or mothers/ or single parent/<br>2. Caregivers/<br>3. Legal Guardians/<br>4. (parent or parents).mp.<br>5. caregiver*.mp.<br>6. carer*.mp.<br>7. guardian*.mp. | 1. Help-seeking Behaviour/<br>2. health information.mp.<br>3. Self-Help Groups/<br>4. self help.mp.<br>5. exp Health Education/<br>6. patient information.mp.<br>7. medical information.mp.<br>8. health information.mp. | 1. Child Health/<br>2. Child/<br>3. Pediatrics/<br>4. (child or children or paediatric* or pediatric*).mp. |
| CINAHL | 1. SU Internet<br>2. forum<br>3. facebook<br>4. social media<br>5. blog*<br>6. Internet | 1. SU (parent* OR father* OR mother* OR single parent*)<br>2. SU caregivers<br>3. SU legal guardians<br>4. (parent OR parents)<br>5. caregiver*<br>6. carer*<br>7. guardian* | 1. SU help seeking behaviour<br>2. health information<br>3. SU self help groups<br>4. self help<br>5. SU health education<br>6. patient information<br>7. medical information<br>8. health information | 1. SU child health<br>2. SU child<br>3. SU paediatrics<br>4. (child OR children OR paediatric* OR pediatric*) |
| Embase | 1. exp Internet/<br>2. forum*.mp.<br>3. facebook.mp.<br>4. social media.mp.<br>5. blog*.mp.<br>6. Internet.mp.<br>7. (web or online).mp | 1. parents/ or fathers/ or mothers/ or single parent/<br>2. Caregivers/<br>3. Legal Guardians/<br>4. (parent or parents).mp.<br>5. caregiver*.mp.<br>6. carer*.mp.<br>7. guardian*.mp | 1. Help-seeking Behaviour/<br>2. health information.mp.<br>3. Self-Help Groups/<br>4. self help.mp.<br>5. exp Health Education/<br>6. patient information.mp.<br>7. medical information.mp.<br>8. health information.mp. | 1. Child Health/<br>2. Child/<br>3. Pediatrics/<br>4. (child or children or paediatric* or pediatric*).mp. |
| PsycINFO | 1. SU Internet<br>2. forum<br>3. facebook<br>4. social media<br>5. blog*<br>6. Internet | 1. SU (parent* OR father* OR mother* OR single parent*)<br>2. SU caregivers<br>3. SU legal guardians<br>4. (parent OR parents)<br>5. caregiver*<br>6. carer*<br>7. guardian* | 1. SU help seeking behaviour<br>2. health information<br>3. SU self help groups<br>4. self help<br>5. SU health education<br>6. patient information<br>7. medical information<br>8. health information | 1. SU child health<br>2. SU child<br>3. SU paediatrics<br>4. (child OR children OR paediatric* OR pediatric*) |

## Table 2  Continued

| Database | Internet search terms | Parent/carer search terms | Use of online health information search terms | Paediatric search terms |
|---|---|---|---|---|
| International Bibliography of Social Science | 1. Ab (Internet* OR forum* OR facebook OR social media OR blog* OR web* OR online) | 1. Ab (parent* OR father* OR mother* OR caregiver* OR guardian* OR carer*) | 1. Ab (help seeking behaviour* OR health information OR self help group* OR self help* OR health education OR patient information OR medical information) | 1. Ab (child health OR child* OR paediatric* OR pediatric*) |

results so were subsequently removed. Others have similarly found subject-specific search strategies rather than methodology-based strategies to be more effective in identifying all relevant studies reporting qualitative findings.[11]

### Study screening methods
One author (BMT) screened the titles and abstracts of all identified papers against the inclusion criteria. Duplicates of papers and those that failed to meet the inclusion criteria were eliminated during this stage. A second author (ET) screened 20% of the titles and abstracts to check for consistency. Full papers for the remaining identified studies were retrieved and assessed for eligibility (see figure 1).

### Quality appraisal and data extraction
One author (BMT) extracted information from the included studies and tabulated it. Assessors (BMT, IM, MS, ET) independently assessed the quality of the included papers, using the Critical Appraisal Skills Programme quality assessment tool for qualitative research.[12] Any discrepancies were discussed with the research team.

### Thematic synthesis
We followed thematic synthesis for the analysis of the included studies,[13] as this method has been successfully used for synthesising qualitative research in systematic reviews in health.[14–19] Thematic synthesis consists of three stages. In stages 1 and 2, we explored and described the data. One author (BMT) read and re-read included papers, then inductively coded each line of the studies' findings according to their content and meaning to develop an initial coding frame. Participants' original data excerpts, as well as the authors' descriptions of findings, were coded in the latest version of NVivo. The coding frame was then discussed and refined within the research team (BMT, ET, IM, AR, NC, MS). In stage 3, we aimed to 'go beyond' individual study findings through developing descriptive themes initially and, subsequently, generating analytical themes.

### Patient and public involvement
We involved an experienced public contributor with experience of eczema (AR) in the aim, design, analysis and write-up stages of the review. AR has a great deal of experience with online support groups, and has often highlighted that the online environment can be very supportive while also potentially bewildering. The contributor's experiences and preferences therefore helped shape the aim and design of the review: to explore parents' experiences of seeking information and support online for long-term physical childhood conditions.

## FINDINGS
### Study selection
The database searches resulted in 2981 records after excluding duplicates. A total of 23[20–42] studies met

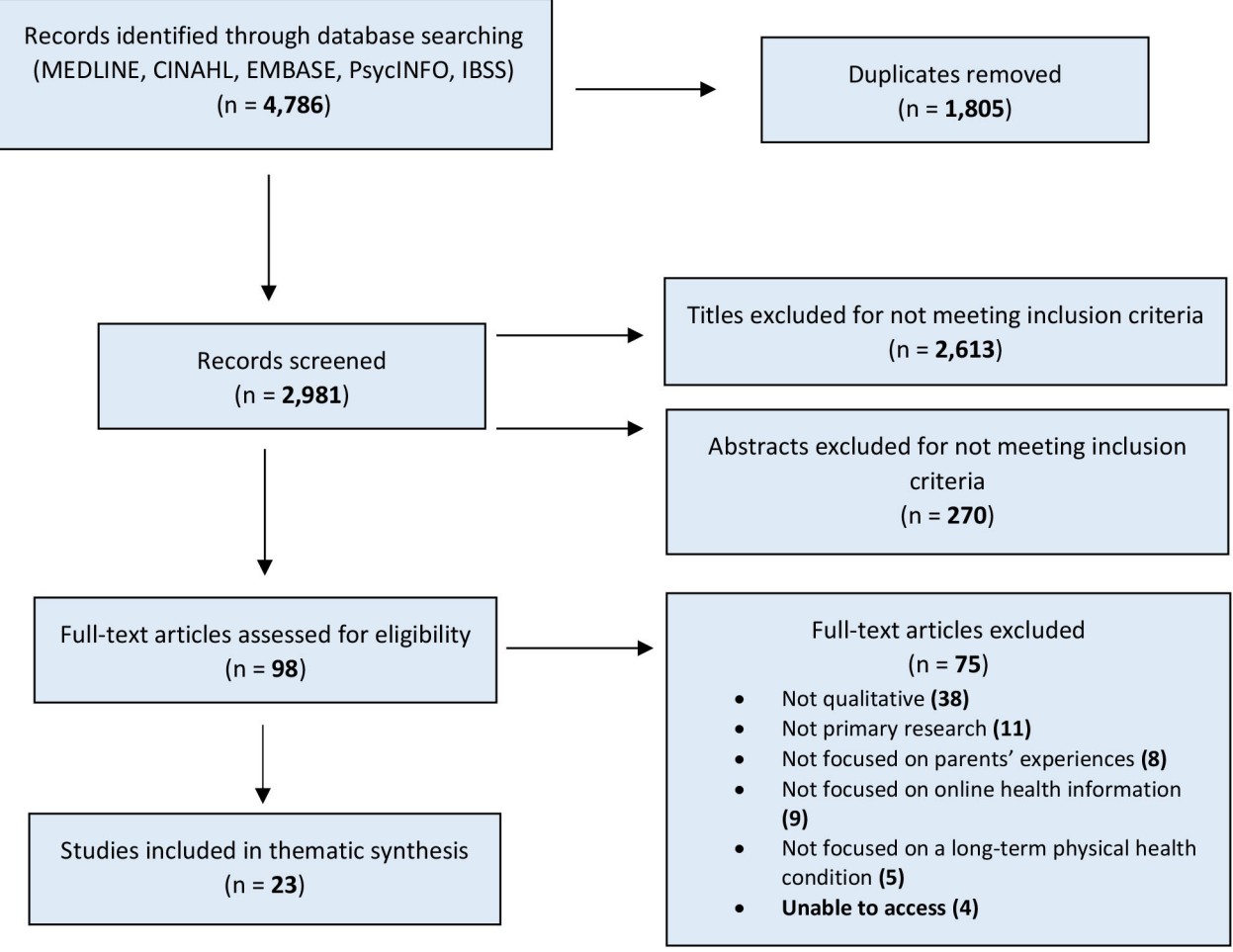

**Figure 1** Preferred Reporting Items for Systematic Reviews and Meta-Analyses diagram of systematic search and study.

all inclusion criteria and were included for thematic synthesis (figure 1).

### Study characteristics
All studies were published in English between December 2004 and July 2019, with the majority of studies (n=14) being published since April 2016. A total of 11 studies were conducted in the Americas (USA,[6] Canada,[4] Brazil[1]), 10 studies were conducted in Europe (UK,[5] Sweden,[2] The Netherlands[2] and Norway[1]) and 2 studies were conducted in Australia. The 23 studies presented the experiences of 559 parents captured through interviews and focus groups, 26 parents through free-text surveys and essays and analyses of 2407 messages that had been posted by parents in online support groups (eg, online forums, social media sites and blogs), though it should be noted that not all studies provided sample size information (table 3).

### Quality of studies
The quality assessment of the included studies revealed that all studies met most quality indicators. All studies had clear statements of the aims of the research, qualitative methodology was appropriate in all studies and

was collected in a way that addressed the aims. We did not exclude any studies based on quality of reporting.

### Synthesis of results
The papers included in this review broadly focused on how parents of children with long-term physical conditions use online resources for health information and support, and what they gain (or not) from using online resources. By synthesising the papers' findings, we interpreted three main analytical themes: (1) developing personal strategies for obtaining and appraising the online world; (2) feeling supported and informed by the online world and (3) perceived risk and concerns about the online world.

Figure 2 illustrates the analytical themes and descriptive themes, and table 4 lists all analytical and descriptive themes, along with references to the relevant papers.

### How do parents develop personal strategies for obtaining and appraising information and support in the online world?
Parents were found to have personal preferences for certain types of websites and their content.[21 23 26–28 31–42] Parents reported often favouring certain websites over others, which included official health information

**Table 3** Summaries of the final 23 papers for thematic synthesis

| Authors of study (country) (ref.) | Aim(s) | Method of data collection (sample size) | Method of data analysis (as written by the authors) | Condition(s) |
|---|---|---|---|---|
| Alsem et al (The Netherlands)[20] | To explore parental and physicians'' experiences with the WWW-roadmap, as well as differences between parents who had used the WWW-roadmap to prepare for a consultation with their rehabilitation physician and parents who had not, in terms of empowerment and self-efficacy, parental and professional satisfaction with the consultation and family centredness of care. | Semistructured interviews (sample size not stated) | Theoretical thematic analysis | Disabilities: cerebral palsy, neuromuscular disorder, metabolic disorder, genetic disorder, spina bifida, various syndromes (eg, Down's), movement disorder, unknown and other \*Included conditions that did not meet inclusion criteria. |
| Benedicta et al (Australia)[21] | To understand how parents search for and appraise the relevance and credibility of online health information regarding their child, and how they use this information in order to help them make better health decisions for their child. | Think aloud task of searching online health info and semistructured interviews (22) | Observation checklist for the think aloud task; inductive thematic analysis for the interviews | 'Subacute' conditions \*Potentially included conditions that did not meet inclusion criteria. Definition of 'subacute' not provided |
| Canty et al (Canada)[22] | To qualitatively and quantitatively assess the use of social media as an information-sharing and support-seeking tool by patients and caregivers. | Analysis of social media postings (sample size not stated) | Modified thematic analysis | Cerebral palsy |
| Gage-Bouchard et al (USA)[23] | To examine the reasons paediatric cancer caregivers engage with cancer-related information on social networking sites and how caregivers appraise and assess the credibility of cancer-related information they encounter on this platform. | Semistructured interviews (40) | Thematic analysis | Cancer |
| Aston et al (UK)[24] | To explore the treatment-related experiences when children and young people take regular prescribed medication. | Semistructured interviews (23 parents) | Thematic analysis | Inpatients taking two or more prescribed medications concurrently at home, prior to admission, for 6 weeks or longer Names of conditions not provided \*Possibly included conditions that did not meet inclusion criteria |
| Hinton et al (UK)[25] | To explore parents' experiences and perspectives of having a baby who needs early abdominal surgery; identify the questions and problems that matter to parents during and after their pregnancy and infant's surgery and identify the long-term impact on parents and families. | Interviews (starting unstructured then semistructured) (44) | Interpretative thematic analysis using a modified grounded theory approach, with constant comparison and exploration of deviant cases | Infants who needed abdominal surgery in the first year of life \*Included conditions that did not meet inclusion criteria |
| Holtz et al (USA)[26] | To develop a website resource, as an initial step in a larger intervention, for parents with a child with type 1 diabetes. | Focus groups (three parents), interviews (3) | Iterative thematic analysis | Type 1 diabetes |
| Perez et al (USA)[27] | Stated as research questions: Research question 1: What types of uncertainty do parents of a child with type 1 diabetes report experiencing? Research question 2: How do parents manage this uncertainty? | In-depth interviews (29) | Thematic analysis | Type 1 diabetes |
| Wright (USA)[28] | To qualitatively explore the informal learning experiences of members of an online social media group hosted by Facebook. | Semistructured interviews (25) and analyses of 604 postings (original and comments) created by members of the online social media | Several grounded theory methodologies | Gastro-oesophageal reflux disease |
| Rehman et al (Canada)[29] | To establish the dominant themes of tweets posted by parents of children diagnosed with cancer and to situate the themes within a theoretical framework that describes the phenomenon of information sharing within a non-work context. | Analysis of the content of 1700 tweets from the Twitter accounts of 15 recruited participants | A deductive subjective interpretation | Cancer |
| Tracey et al (Australia)[30] | To explore how parents acquire information to enhance their understanding of their child's disability and individualised funding schemes. | Focus groups (56 parents, 13 groups) | A two stage, mixed method sequential approach which followed an initial survey | 'Disability' \*Possibly included conditions that did not meet inclusion criteria |

Continued

**Table 3** Continued

| Authors of study (country) (ref.) | Aim(s) | Method of data collection (sample size) | Method of data analysis (as written by the authors) | Condition(s) |
|---|---|---|---|---|
| Alsem et al (The Netherlands)[31] | To explore the experiences of Dutch parents of children with disabilities as regards their information needs, and to describe the process of seeking and evaluating information and the different sources of information. | Semistructured interviews (15) | Open coding thematic analysis | Cerebral palsy, gross motor function classification system 2–5, rare disorders (eg, genetic syndromes and metabolic and mitochondrial diseases) *Possibly included conditions that did not meet inclusion criteria |
| Kirk and Milnes (UK)[32] | To explore how online peer support is used by young people and parents to support self-care in relation to cystic fibrosis. | Analysis of discussion group postings (103) | Inductive grounded theory | Cystic fibrosis |
| Vogel et al (Canada)[33] | To explore parental usage of the National Kidney Foundation website, track visitor behaviour, evaluate usability and design, establish ways to improve user experience and identify ways to redesign the website. | Mixed methods: Google analytic usage reports and focus groups (4) | Inductive content analysis and constant comparative approach | A condition requiring neurosurgery |
| Santer et al (UK)[34] | To explore parents and carers' experiences of searching for information about childhood eczema on the internet. | Semistructured interviews (31) | Content analysis and constant comparative approach | Eczema |
| Cargnin Pimentel et al (Brazil)[35] | To describe the experience of patients with cystic fibrosis and their families using the internet as a means for information, interaction and exchange of experiences with this disease. | Questionnaire (7) | Content analysis | Cystic fibrosis |
| Nordfeldt et al (Sweden)[36] | To explore parents' views on their information seeking, internet use and social networking online, and to identify implications for the future development of internet use in this respect. | Focus groups (27) | Inductive qualitative content analysis | Type 1 diabetes |
| Gage and Panagakis (USA)[37] | To examine how parents think about, evaluate, access and use the internet to seek information related to their child's cancer. | Mixed methods: survey and structured interviews (41) | Grounded theory | Cancer |
| Gundersen (Norway)[38] | To explore how the internet can function as a resource that parents can draw on to cope with their situation. | Semistructured interviews (10) | Not a specified method but looks like thematic analysis with comparative approach based on theorist | Rare genetic disorders that entail developmental and/or physical problems (eg, allergies and epilepsy, and two had severe physical and learning problems) *Included conditions that did not meet inclusion criteria |
| Stewart et al (Canada)[39] | To gather parents' preferences of support to design an innovative online peer support and test it. | Semistructured interviews (44) | Deductive thematic content analysis | Asthma and allergies |
| Nordfeldt et al (Sweden)[40] | To explore patients' and parents' attitudes toward a local Web 2.0 Portal tailored to young patients with type 1 diabetes and their parents. | Essays (19) | Content analysis | Type 1 diabetes |
| Roche and Skinner (USA)[41] | To describe how parents of a child referred for genetic services searched the internet for information, summarise how they interpreted and evaluated the information they obtained and identify barriers that they encountered. | Semistructured interviews (100) | Constant comparative method | Genetic |
| Nettleton et al (UK)[42] | To explore parents and children's experiences of using the internet for chronic childhood conditions. | Qualitative interviews (69) | Strategic comparisons approach | Eczema, asthma and diabetes |

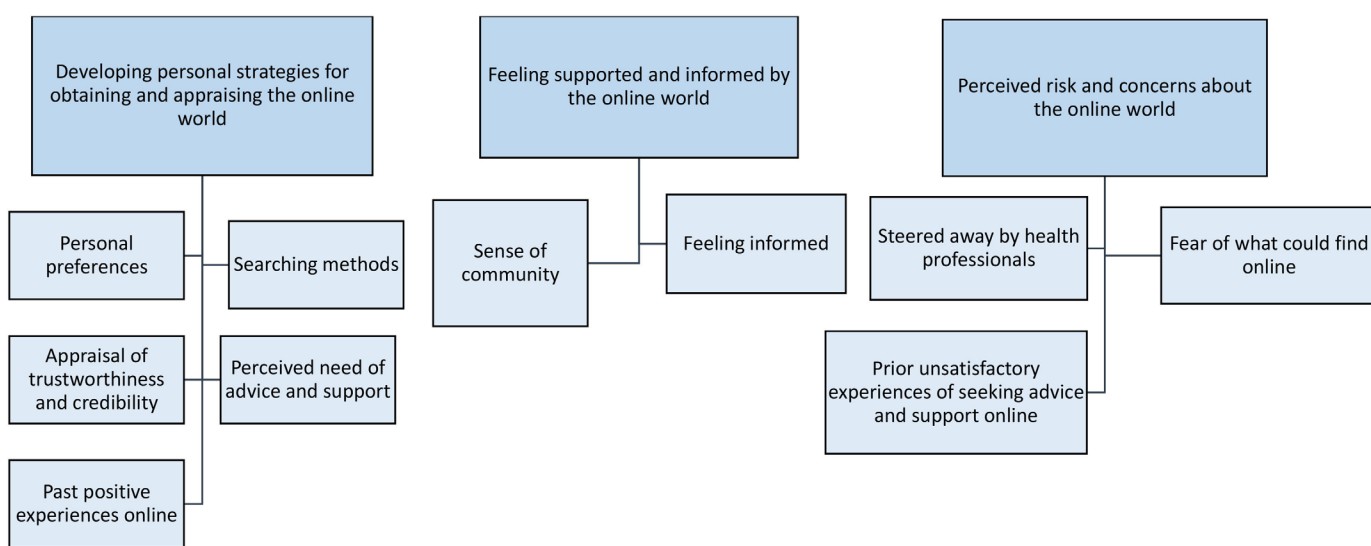

**Figure 2** Illustration of the developed analytical themes and descriptive themes.

sites, charity websites, research sites, online discussion forums and social media sites. From within the different types of websites, parents also expressed informational preferences including medically moderated content, personalised advice and content

dedicated to the condition without advertisements and other irrelevances. Parents also expressed functional preferences for online resources which included closed/private communities, live chat rooms and notifications for new content. 'For me, this is a

| Table 4 | Overview of analytical and descriptive themes across the presenting papers | |
| --- | --- | --- |
| **Analytical theme** | **Descriptive theme** | **Paper references (total number)** |
| Developing personal strategies for obtaining and appraising the online world | | |
| | Personal preferences | 22, 24, 27, 28, 29, 32, 33, 34, 35, 36, 37, 38, 39, 40, 41, 42, 43 (17) |
| | Searching methods | 22, 24, 29, 31, 32, 34, 35, 37, 38, 39, 42, 43 (12) |
| | Appraisal of trustworthiness and credibility | 22, 24, 27, 28, 29, 31, 32, 34, 35, 36, 37, 38, 43 (13) |
| | Perceived need of information and support | 21, 22, 23, 26, 27, 28, 29, 33, 34, 35, 36, 37, 38, 39, 40, 41, 42, 43 (18) |
| | Past positive experiences online | 21, 22, 24, 27, 28, 29, 31, 35, 36, 37, 38, 39, 40, 41, 42, 43 (16) |
| Feeling supported and informed by the online world | | |
| | Sense of community | 23, 24, 25, 26, 27, 28, 29, 30, 33, 35, 36, 37, 38, 39, 40, 43 (16) |
| | Feeling informed | 21, 22, 23, 24, 25, 26, 27, 29, 30, 32, 33, 36, 37, 38, 39, 40, 41, 42, 43 (19) |
| Concerns and perceived risks of the online world | | |
| | Prior unsatisfactory experiences of seeking information and support online | 21, 22, 25, 28, 29, 31, 32, 34, 35, 36, 27, 38, 39, 41, 42, 43 (16) |
| | Steered away by health professionals | 22, 26, 29, 32, 34, 38 (6) |
| | Fear of what could find out online | 22, 25, 28, 29, 34, 38, 39, 42 (8) |

funnel. This group funnels you. People can get quick answers, but if you want more in-depth information, we always do the link to a website because it has way more in-depth information than what you can get in the group online.'[28] (Gastro-oesophageal reflux disease, interview and online forum study)

Parents also described a variety of searching methods which enabled them to obtain online information and support.[21 23 28 30 31 33 34 36–38 41 42] Some visited the internet with a specific question in mind, others reported typing in keywords that they had heard during their medical consultations and others posted questions in a variety of online resources and combined the information with advice received offline. 'The same question on WhatsApp, Facebook and at the school.'[31] (Various conditions including cystic fibrosis, metabolic conditions and other disabilities, interview study)

Parents also reported appraising the trustworthiness and credibility of online resources, which influenced whether they decided to use the online resource.[21 23 26–28 30 31 33–37 42] Some had reported more specific tactics such as matching up information from a variety of websites, checking the affiliation/source of the website or the background of individual posters, the possible sales and marketing intentions behind the websites and comparing the advice with existing knowledge, whereas others implied that the reliability of online resources had been at least somewhat on their mind. '[The website] that came up first, they've probably got advertisements in there… everything is paid, so I am a bit sus on that.'[21] (Subacute conditions, think aloud and interview study)

Parents' perceived needs for information and support also appeared to influence whether and how they searched for information and support online.[20–22 25–28 32–42] This involved the present stage and severity of their child's condition, such as a recent diagnosis or a change in symptoms, and an unsatisfying recent medical consultation. 'Five minutes after we got the diagnosis… we got on the computer. I didn't know what else to do.'[41] (Genetic conditions, interview study)

Finally, parents also expressed past positive experiences with the online world, and how these often encouraged continuous use.[20 21 23 26–28 30 34–42] Experiences included the internet's convenient and accessible nature for obtaining advice any time of the day, through a variety of mediums, for exploring sensitive topics and within the protection of anonymity and general successes of obtaining desired information. 'You know when you post even though it's like 5:00 in the morning, you've got other people that are also awake and even though it is 5:00 in the morning, they respond as well within a timely manner.'[28] (Gastro-oesophageal reflux disease, interview and online forum study)

### How and why do parents feel supported and informed by the online world?

The sense of community from using social media sites, discussion forums and blogs for peer support was also emphasised.[22–29 32 34–39 42] Connecting with other parents with similar experiences online was described as comforting, and often reduced their loneliness and sorrow. The importance of building trust and relationships in online communities was also reported, which enabled parents to gain more advice from one another, to share common language, to be treated as equals and to constantly be there to support one another. Parents also demonstrated that exchanging experiences online helped them to feel more reassured and validated about their situations. Also, parents expressed how online information and support that derives from others' experiences is invaluable, which often lacks in health professionals' advice.

> …I'm so happy that I found this page. For me to share & read about that there are other parents in my shoes. It breaks my heart watching my little girl try so hard to walk & play with other kids… Her condition has really taken a toll on me. Physically, mentally, & emotionally. I cry a lot. I'm just relieved there's parents on here that I can open up to.[22] (Cerebral palsy, online forum study)

Parents across nearly all papers emphasised feeling informed through accessing and using information and support obtained online.[20–26 28 29 31 32 35–42] The sense of feeling better informed was reported to derive from parents' development of condition-related knowledge including information about diagnosis, prognosis, triggers and treatments. Parents often reported that their increase in medical knowledge helped them to feel in more control of managing their children's condition and to cope with stress and during uncertain times such as changes in symptoms and treatments. Accessing information online was also reported to facilitate parents in subsequent medical consultations, through helping them to feel more equipped to understand what their health professionals were telling them, to structure questions and to recommend plans for their children's treatments. Parents also reported feeling good about sharing their own experiences online about managing their children's conditions, such as with treatments, consultations and coping methods. 'I think it allows you to then make a decision on how serious something might be or whether you need to take action straight away or if it's something that you can potentially wait up and see what happens.'[21] (Subacute conditions, think aloud and interview study)

### How and why do parents develop concerns and perceive potential risks from seeking information and support online?

While less common than positive accounts of feeling supported and informed, some parents reported the negative side to information and support online. Some parents reported that prior unsatisfactory experiences of seeking information and support online had discouraged them from using online resources.[20 21 24 26–28 30 31 33–35 37 38 40–42] Previous experiences included not being able to find their desired information and support, being overwhelmed by

the volume of content online, feeling that the internet does not replace real people and a lack of digital literacy skills and experience. 'I choose not to (go online) because there are so many diverse opinions out there. There's so much garbage out there.'[37] (Cancer, survey and interview study)

Parents also reported experiences of being steered away by health professionals from using the internet for information and support.[21 25 28 31 33 37] This was predominantly in papers that focused more on life-limiting conditions (eg, needing surgery in first year of life, gastro-oesophageal reflux disease, cerebral palsy, neurological, cancer).

> (Physician B) was very adamant, 'Don't you dare touch that internet, do not look at it, do not—you listen to what I say, I'm the boss, and this is the way it's going to run… I just wanted the definition … I just wanted to know what the words meant.…'[33] (Conditions requiring neurosurgery, focus group study)

Finally, some parents reported that the fear of what they could find out online prevented them from using online resources for information and support.[21 24 27 28 33 37 38 41] Some reported having read distressing stories in the past, and others simply did not want to risk uncovering distressing information that they could relate to. These papers also focused more on life-limiting conditions (cancer, type 1 diabetes, rare genetic conditions, neurological).

> I was careful with going online because I think the internet can be a really scary place. We got into a couple of those forums that tend to be sort of doom and gloom and focus on all the negative things. It's all the people that are struggling the most that seem to post in some of those forums.[27] (Type 1 diabetes, interview study)

## DISCUSSION
### Summary of findings
This systematic review and thematic synthesis revealed that parents developed a variety of strategies to obtain and appraise information and support online in the context of caring for long-term physical childhood conditions. These processes were based on their personal preferences, their search methods, the outcome of their appraisal of trustworthiness, the degree of their perceived need for information and support and previous positive experiences online. In most cases, parents reported going on to experience the benefits of accessing information and support online. This included the sense of community gained through connecting with others in a similar situation online, building relationships and trust and feeling reassured and validated, which was available from online support communities especially. Parents also reported feeling better informed about their children's conditions, which helped them feel more in control in subsequent medical consultations and felt good about

helping similar others by sharing their own experiences. However, some parents also emphasised the concerns and perceived risks of obtaining and using information and support online. These often stemmed from prior unsuccessful online experiences, having been told to avoid the internet by health professionals, and due to the fear of the distressing stories that they might come across online.

### Findings in context of existing research
The findings from this study are largely consistent with other reviews that have explored peoples' experiences with health information online,[5 6] although these reviews had not focused specifically on the online experiences of parents of children with long-term physical conditions. However, our review uncovered more about the initial processes through which parents developed search strategies to obtain and evaluate health information and support online and the motivations behind these, which lead to the subsequent positive and negative experiences identified. Our review also drew more attention to the negative experiences that parents experienced online compared with these reviews,[5 6] which could be due to the additional challenges of caring for a child rather than self-caring.

Collating and synthesising the experiences of parents of children with a variety of long-term conditions provided additional insights into parents' experiences with the online world. Drawing on the different experiences discussed suggested that the potential harms of the online world were most apparent for parents of children with life-limiting conditions, or whose conditions were unstable (eg, cerebral palsy, cancer, type 1 diabetes, rare genetic conditions, gastro-oesophageal reflux disease). Concerns and perceived risks were not found in the papers focusing on eczema, asthma and other allergies and other disabilities. This suggests that parents of children with life-limiting or unstable conditions use information and support online with more caution.

This review was also able to further identify that the types of information and support sought by parents differed according to the stage and severity of their children's conditions. The need for condition-related information was again more apparent across the papers that focused on the life-limiting or unstable conditions, whereas the need for peer support was apparent across all stages and severities (life-limiting, uncertain to stable and well managed). This suggests that parents' need for condition-specific information comes first and is not needed in long term, whereas the need for peer support is continuous.

### Strengths and limitations
This is the first systematic review to synthesise the existing qualitative literature exploring parents' experiences of seeking health information and support online for long-term physical childhood conditions, providing new insights and developing the research field in this area.

A limitation to this study was that some of the included studies for synthesis also explored parents' online experiences for conditions that are not long term and physical (eg, acute conditions, disabilities of learning, language and cognition, mental health conditions and health promotion in childhood), within the same paper. Some of the findings in these papers were not separated according to different conditions, therefore this review fails to identify possible differences between the online experiences of parents of children with long-term physical conditions and other types of conditions. However, this was never an aim of the study. Some included papers had limitations in reporting of methodological detail, for instance, sample size information. Additionally, included studies typically presented limited demographic data so it was not possible to explore how such factors may affect parents' experiences in the online world. We also excluded quantitative studies which could have been used to support the qualitative findings, and did not search the grey literature which could have excluded some relevant studies. Finally, we analysed only the results sections of each paper, meaning that broader contextual factors that may have arisen in the discussion sections were not included although the discussion sections were read and checked in case.

### Implications and future research

Improved understanding of parents' experiences of seeking health information and support online for long-term physical childhood conditions may help health professionals treating these families. The findings from this study could be used to inform health professionals about parents' experiences online: such as what they want from information and support online, why they choose online resources, what they are influenced by and the types of information and support that they access online.

Parents' preferences and perceived benefits of online resources could be useful for future studies designing online resources for parents of children with long-term conditions. Specific examples that this study identified were online resources that contain medically moderated content, reminders to use online resources, frequently active members to reply to posts, easy to use and information tailored to the child's condition and its severity. A future systematic review could collate which online resources for certain conditions are evidence based and meet the needs of parents, so that health professionals can signpost patients to such online resources.

### CONCLUSION

This systematic review and synthesis of qualitative literature comprehensively explored parents' experiences with health information online for long-term physical childhood conditions. This highlights the support that parents receive from online resources, but also their perceptions of potential risks and the processes by which they attempt to establish trustworthiness of sources. These findings could help health professionals to understand parents'

perspectives about information and support online, and potentially help them identify robust resources that also meet patients' needs.

**Author affiliations**
[1]Primary Care, Population Sciences and Medical Education, Faculty of Medicine, University of Southampton, Southampton, UK
[2]Centre of Evidence-Based Dermatology, Faculty of Medicine and Health Sciences, University of Nottingham, Nottingham, UK
[3]Division of Rehabilitation, Ageing and Wellbeing, Faculty of Medicine and Health Sciences, University of Nottingham, Nottingham, UK

**Acknowledgements** We would like to thank Paula Sands for her expertise in developing the search strategy for this study.

**Contributors** BMT developed and refined the protocol, undertook the systematic review, carried out data extraction, quality assessment and thematic synthesis and drafted the manuscript. ET refined the protocol, carried out eligibility screening process, quality assessment and thematic synthesis and wrote the manuscript. MS and IM refined the protocol, carried out quality assessment and thematic synthesis and wrote the manuscript. AR and NC refined the protocol, contributed to interpretation of findings and wrote the manuscript. All authors read and approved the final manuscript.

**Funding** This study was part of a PhD studentship funded by the National Institute for Health Research (NIHR) School for Primary Care Research. The views expressed are those of the authors and not necessarily those of the NIHR or the Department of Health and Social Care.

**Competing interests** None declared.

**Patient consent for publication** Not required.

**Provenance and peer review** Not commissioned; externally peer reviewed.

**Data availability statement** No data are available.

**ORCID iDs**
Bethan Mair Treadgold http://orcid.org/0000-0002-0255-7422
Emma Teasdale http://orcid.org/0000-0001-9147-193X
Ingrid Muller http://orcid.org/0000-0001-9341-6133

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
