## [Reviewer comments · BMJ Open]

ARTICLE DETAILS

TITLE (PROVISIONAL)	Parents and carers' experiences of seeking health information and support online for long-term physical childhood conditions: A systematic review and thematic synthesis of qualitative research
AUTHORS	Treadgold, Bethan; Teasdale, Emma; Muller, Ingrid; Roberts, Amanda; Coulson, Neil; Santer, Miriam

VERSION 1 – REVIEW

REVIEWER	Dr Lisa Arai Solent University, Southampton, UK
REVIEW RETURNED	01-Aug-2020

GENERAL COMMENTS	This is a clearly written and engaging paper on parents' use of online information. Three observations are made here. First, you write that: 'We involved an experienced public contributor with experience of eczema (AR) in the analysis and write-up stages of the review', but no further information is provided about the role of this person in the research process. Second, the inclusion of papers focusing on a wider range of children's conditions than just physical ones is a significant (and unavoidable) limitation (which you acknowledge on p. 19). Can you quantify the extent of this, or make it clearer which studies collected data from parents of children with conditions not meeting inclusion criteria (in table 3 maybe)? Third, there were one or two instances of 'clunky' terminology ('We sought studies that explored the experiences of parents of children with any long-term physical condition(s) around using the Internet for information and support' and 'although this was not ever an aim of the study' - consider rewording).
--

REVIEWER	Helen Roberts UCL Great Ormond Street Institute of Child Health UK
REVIEW RETURNED	03-Aug-2020

GENERAL COMMENTS	Thank you for sending me this interesting and well-conducted review. I have only minor observations listed in the order in which they appear rather than order of importance. 1. PRISMA- the authors implicitly refer to risks of bias and assumptions made re benefits of harms - so I'm not sure that these merit the n/a. These issues are just as important for qualitative studies than for any other. 2. ENTREQ -online support groups not mentioned in the search strategy - presumably because they are embedded in other studies ? 3. ABSTRACT: The part of the abstract starting 'in most cases' etc is more discussion than results. I wasn't sure about the 'feeling
---

	empowered by further medical knowledge' on the basis of the data shown - I felt that a more balanced analysis of benefits and harms is made at the end of the paper itself. 4. 'Convenient' for health professionals seems a rather odd use of the term. 5. 'Reviews' appears under exclusions. Does this mean that there were no reviews of reviews? 6. What is an 'essay survey' ? [5 and 6 above can be responded to by reference to the studies - but if space allows, useful in the article itself] 7. You mention lack of sample size for some studies as a limitation - might it have been an exclusion criterion ? how can you be sure that quotations not cherry picked ? (I know that this can be an irritating Q for qualitative scholars but as a common one, worth considering). 8. In terms of presentation, I recommend starting with 'sense of community' which is what comes over most strongly in your results, and putting closer together those relating to 'almost all felt empowered' with those who did not but referred to risks. 9. 'Empowerment' is a term which we social scientists tend to like, but I wonder how much it resonates with parents - it rarely appears in parental accounts but more often in analyses. What parents seem to be talking about more frequently is an increased sense of control (though that isn't quite the right term either). As parents or carers may also be readers of this piece, it's more than usually important for it to be in plain English. 10. In the interests of transparency, where you have a quotation, you may want to consider putting into the article (as well as the reference) details of whether this was an online support group or a qualitative study.
--	--

VERSION 1 – AUTHOR RESPONSE

Reviewer comments	Author response
Reviewer 1	Thank you for your comments.
1. You write that: 'We involved an experienced public contributor with experience of eczema (AR) in the analysis and write-up stages of the review', but no further information is provided about the role of this person in the research process.	Thank you for the opportunity to clarify. AR was involved in the analysis and write-up of the review. However, AR has been involved in the wider programme of my research (PhD exploring parents' experiences with advice and support about eczema in the online world). She helps run the Nottingham Support Group for carers of children with eczema which has an online presence of over 6500 followers (@eczemasupport). We have added some more details to the Patient and Public Involvement section: 'We involved an experienced public contributor with experience of eczema (AR) in the aim, design, analysis and write-up stages of the review. AR has a great deal of

	experience with online support groups, and has often highlighted that the online environment can be very supportive whilst also potentially bewildering. The contributor's experiences and preferences therefore helped shape the aim and design of the review: to explore parents' experiences of seeking information and support online for long-term physical childhood conditions.' (p.8).
2. The inclusion of papers focusing on a wider range of children's conditions than just physical ones is a significant (and unavoidable) limitation (which you acknowledge on p. 19). Can you quantify the extent of this, or make it clearer which studies collected data from parents of children with conditions not meeting inclusion criteria (in table 3 maybe)?	Thank you for your comment. We have amended Table 3 by adding an asterisk with a note within the condition(s) column for every paper that included children's conditions that did not meet our inclusion criteria. E.g. * 'Included conditions that did not meet inclusion criteria'. Or * 'Possibly included conditions that did not meet inclusion criteria' for when the authors have put 'other conditions' and not specified what they are.
3. There were one or two instances of 'clunky' terminology ('We sought studies that explored the experiences of parents of children with any long-term physical condition(s) around using the Internet for information and support' and 'although this was not ever an aim of the study' - consider rewording).	Thank you. Both instances have been reworded: 1) 'We sought studies that explored the experiences of parents of children with any long-term physical condition(s) around using the Internet for information and support (Table 1).' Has been changed to: 'We sought studies that explored parents' experiences of using the Internet for information and support about long-term physical childhood condition(s) (Table 1).' (p.4). 2) 'Some of the findings in these papers were not separated according to different conditions, therefore this review fails to identify possible differences between the online experiences of parents of children with long-term physical conditions, and other types of conditions, although this was not ever an aim of the study.' Has been changed to: 'Some of the findings in these papers were not separated according to different conditions, therefore this review fails to identify possible differences between the

	online experiences of parents of children with long-term physical conditions, and other types of conditions. However, this was never an aim of the study.' (p.20).
Reviewer 2	Thank you for your comments.
1. PRISMA- the authors implicitly refer to risks of bias and assumptions made re benefits of harms - so I'm not sure that these merit the n/a. These issues are just as important for qualitative studies than for any other.	Thank you for your comment. We have amended the PRISMA checklist to reflect where risk of bias has been discussed in the manuscript.
2. ENTREQ -online support groups not mentioned in the search strategy - presumably because they are embedded in other studies ?	Thank you. We used the phrase 'online support groups' as an umbrella term to cover the different types of online support groups (online discussion forums, social media sites and online blogs), which were individually included in the search strategy. We worked with a medical librarian on this search strategy. To clarify the umbrella term 'online support groups', we have amended where the phrase is written in text: 'The 23 studies presented the experiences of 559 parents captured through interviews and focus groups, 26 parents through open-ended surveys or essay surveys, and analyses of 2407 messages that had been posted by parents in online support groups though it should be noted that not all studies provided sample size information (Table 3).' (p. 9). Has been changed to: 'The 23 studies presented the experiences of 559 parents captured through interviews and focus groups, 26 parents through free-text surveys and essays, and analyses of 2407 messages that had been posted by parents in online support groups (e.g. online forums, social media sites and blogs) though it should be noted that not all studies provided sample size information (Table 3).' (p. 9). We have also added to the manuscript that we worked with a medical librarian and have included this in the acknowledgments: 'A comprehensive search strategy was developed with a medical librarian from the University of Southampton for each database (Table 2)'. (p.5)

	'Acknowledgements We would like to thank Paula Sands for her expertise in developing the search strategy for this study.' (p.26)
3. ABSTRACT: The part of the abstract starting 'in most cases' etc is more discussion than results. I wasn't sure about the 'feeling empowered by further medical knowledge' on the basis of the data shown - I felt that a more balanced analysis of benefits and harms is made at the end of the paper itself.	Thank you for your comment. We have amended the abstract in line with your feedback: 'In most cases, they sought the benefits of online information and support, which included feeling empowered with further medical knowledge and perceived control, reassurance and validation from online communities.' Has been changed to: 'Many parents sought the benefits of online information and support, which included reassurance and validation from online communities, and feeling they had greater knowledge about their children's conditions.' (p.2)
4. 'Convenient' for health professionals seems a rather odd use of the term.	Thank you, the term has been replaced: 'Convenient for health professionals and researchers interacting with this population' Has been changed to: 'Useful for health professionals and researchers interacting with this population' (p.3).
5. 'Reviews' appears under exclusions. Does this mean that there were no reviews of reviews?	Thank you for the opportunity to clarify. Yes, we sought primary research papers only, therefore no reviews were included the synthesis. This is specified in the eligibility criteria in Table 1 (p.5) and in the abstract.
6. What is an 'essay survey' ? [5 and 6 above can be responded to by reference to the studies - but if space allows, useful in the article itself]	Thank you. Apologies that this was not clearly written. This was a method that one of the included studies used who asked participants to write an essay (similarly to an open-ended survey with a large free-text box). We have rephrased as: 'free-text surveys and essays' (p.2). 'free-text surveys or essays' (p.8)
7. You mention lack of sample size for some studies as a limitation - might it have been an exclusion criterion ? how can you be sure that quotations not cherry picked ? (I know that this can be an irritating Q for qualitative scholars but as a common one, worth considering).	Thank you for your comment. We agree that this is a limitation of the study. We did not consider making this an exclusion criteria at the time however we will bear this in mind for future research, thank you.

	This has been added to the limitations section of the manuscript: 'Some included papers had limitations in reporting of methodological detail, for instance sample size information.' (p.20)
8. In terms of presentation, I recommend starting with 'sense of community' which is what comes over most strongly in your results, and putting closer together those relating to 'almost all felt empowered' with those who did not but referred to risks.	Thank you for your recommendation. We have amended the manuscript so that this sub-theme is reported first in every instance when discussing the analytical theme 'How and why do parents feel supported and informed by the online world?'. We hope that the reviewer agrees it is logical that how parents search and appraise online information' appears prior to this as this is the start of parents' online journey
9. 'Empowerment' is a term which we social scientists tend to like, but I wonder how much it resonates with parents - it rarely appears in parental accounts but more often in analyses. What parents seem to be talking about more frequently is an increased sense of control (though that isn't quite the right term either). As parents or carers may also be readers of this piece, it's more than usually important for it to be in plain English.	Thank you for this very interesting insight. We have renamed this sub-theme as: 'Feeling informed' (e.g. p.15). And rephrased relating sentences to: 'Feeling better informed about their children's conditions' (e.g. p.17).
10. In the interests of transparency, where you have a quotation, you may want to consider putting into the article (as well as the reference) details of whether this was an online support group or a qualitative study.	Thank you for your suggestion. We have amended the labels for all quotes to include the type of study. E.g. '(29; Gastro-oesophageal reflux disease, Interview and online forum study)'

VERSION 2 – REVIEW

REVIEWER	Dr Lisa Arai Solent University, Southampton, UK
REVIEW RETURNED	16-Oct-2020
GENERAL COMMENTS	clearly written and interesting paper that reports interesting findings.
REVIEWER	Helen Roberts UCL Gt Ormond Street Institute of Child Health
REVIEW RETURNED	15-Oct-2020
GENERAL COMMENTS	Thank you for your responses. This psper provides some useful insights. Thank you.

	I wonder though whether librarians get their fair share of authorship credit given the substantial contribution they make to review papers - obviously a decision for you and the librarian !
--	---

VERSION 2 – AUTHOR RESPONSE

Please find our responses to the minor revisions below:

Comments	Author response
Please add months to the years where dates are mentioned in the manuscript.	Thank you for your comment. We have added months to the years: “All studies were published in English between December 2004 and July 2019, with the majority of studies (n = 14) published since April 2016.” (page 9 of marked copy; page 8 of clean copy).
Strengths and limitations (Article Summary) – Please revise as currently it seems to focus more on the results - add up to 5 single sentence summary points that focus on the methods alone.	Thank you for your comment. We have revised the article summary to focus on the methods:  • “The first systematic review to synthesise the existing qualitative literature exploring parents/carers’ experiences of seeking health information and support online for long-term physical childhood conditions • A comprehensive study synthesising the experiences of hundreds of parents/carers of children with a broad range of long-term physical conditions which provides new insights and develops the research field • A systematic and comprehensive search of the literature using multiple databases coupled with detailed synthesis of included studies • Excludes the online experiences of parents/carers of children with

	acute, mental health, or developmental conditions, and otherwise healthy children  • Excluded quantitative studies which could have been used to support the qualitative findings” (page 3)
Thank you for your responses. This paper provides some useful insights. Thank you. I wonder though whether librarians get their fair share of authorship credit given the substantial contribution they make to review papers - obviously a decision for you and the librarian !	Thank you for your kind comment. We agree that librarians often provide a substantial contribution to obtaining relevant papers. In line with the BMJ policy and ICMJE recommendations for authorship, we have decided not to credit authorship to the librarian who helped develop the database search strategy for our review, because she was not involved in any other areas of the work. We have included her in the acknowledgements.
A clearly written and interesting paper that reports interesting findings.	Thank you for your kind comment.